# Interprofessional Approaches to the Treatment of Mild Traumatic Brain Injury: A Literature Review and Conceptual Framework Informed by 94 Professional Interviews

**DOI:** 10.3390/medsci13030082

**Published:** 2025-06-23

**Authors:** John F. Shelley-Tremblay, Teri Lawton

**Affiliations:** 1Department of Psychology, University of South Alabama, Mobile, AL 36688, USA; 2Perception Dynamics Institute, PO Box 231305, Encinitas, CA 92023-1305, USA; pathtoreading1@gmail.com

**Keywords:** mild traumatic brain injury, concussion, interdisciplinary care, post-concussion symptoms, neurorehabilitation, stakeholder interviews, care coordination, implementation science, team-based care, patient-centered care

## Abstract

Background/Objectives: Mild traumatic brain injury (mTBI) presents with persistent, heterogeneous symptoms requiring multifaceted care. Although interdisciplinary rehabilitation is increasingly recommended, implementation remains inconsistent. This study aimed to synthesize existing literature and clinician perspectives to construct a practice-informed conceptual framework for interprofessional mTBI rehabilitation. Methods: Structured interviews were conducted with 94 clinicians—including neurologists, neuropsychologists, optometrists, occupational and physical therapists, speech-language pathologists, neurosurgeons, and case managers—across academic, private, and community settings in the United States. Interviews followed a semi-structured format adapted for the NIH I-Corps program and were analyzed thematically alongside existing rehabilitation literature. Results: Clinicians expressed strong consensus on the value of function-oriented, patient-centered care. Key themes included the prevalence of persistent cognitive and visual symptoms, emphasis on real-world goal setting, and barriers such as fragmented communication, reimbursement restrictions, and referral delays. Disciplinary differences were noted in perceptions of symptom persistence and professional roles. Rehabilitation technologies were inconsistently adopted due to financial, training, and interoperability barriers. Equity issues included geographic and insurance-based disparities. A four-domain conceptual framework emerged: discipline-specific expertise, coordinated training, technological integration, and care infrastructure, all shaped by systemic limitations. Conclusions: Despite widespread clinician endorsement of interprofessional mTBI care, structural barriers hinder consistent implementation. Targeted reforms—such as embedding interdisciplinary models in clinical education, expanding access to integrated technology, and improving reimbursement mechanisms—may enhance care delivery. The resulting framework provides a foundation for scalable, patient-centered rehabilitation models in diverse settings.

## 1. Introduction

Mild traumatic brain injury (The term ‘mTBI’ is used in this article to include cases commonly referred to as concussion, recognizing that terminology varies across disciplines) (mTBI), often synonymous with concussion, represents a significant and growing public health concern, affecting millions annually across the globe. Although most individuals recover within weeks, a substantial minority experience persists post-concussion symptoms (PPCSs) that extend beyond the expected recovery window and profoundly impact the quality of life, occupational function, and psychosocial well-being [1]. Persistent symptoms are often heterogeneous and multifactorial, necessitating nuanced, flexible, and interdisciplinary models of care [2].

Clinical guidelines increasingly call for team-based, patient-centered approaches to PPCS, in which professionals from different disciplines coordinate to provide comprehensive rehabilitation. Yet, despite this consensus, implementation across systems remains inconsistent, with few studies exploring the dynamics of interprofessional collaboration in real-world contexts. This article reviews the literature on interdisciplinary rehabilitation for mTBI and synthesizes insights from 94 interviews with rehabilitation professionals—including neurologists, neuropsychologists, optometrists, speech-language pathologists, occupational and physical therapists, and others—to glean practical lessons and conceptual insights.

This investigation emerged from a Phase I NINDS-funded clinical trial conducted by the Perception Dynamics Institute, which sought to evaluate a novel intervention targeting dorsal stream visual processing deficits in individuals with mTBI. As part of the NIH I-Corps program—an initiative that supports the commercialization of academic research and the identification of unmet clinical needs—our team conducted over 100 in-depth interviews with potential stakeholders, including healthcare professionals from a diverse array of rehabilitation disciplines. The team consisted of the first author and second author. The I-Corps program emphasizes experiential learning and real-world discovery; it requires participants to validate the translational potential of their interventions through structured stakeholder engagement. Through this process, we gathered first-hand insights into the diagnostic and therapeutic challenges of mTBI, especially those related to fragmented care, under-recognized visual impairments, and the lack of standardized interprofessional collaboration.

These interviews offered an unexpectedly rich source of qualitative data, revealing consistent patterns in how rehabilitation professionals—neurologists, neuropsychologists, optometrists, speech-language pathologists, occupational and physical therapists, and others—conceptualize and coordinate care for individuals with mTBI. The convergence of their experiences highlighted both the promise and the pitfalls of interdisciplinary practice in brain injury rehabilitation. Accordingly, this article draws on those interviews to review the existing literature on interdisciplinary rehabilitation and to construct a conceptual framework grounded in real-world clinical practice. It aims to illuminate emerging trends, persistent barriers, and opportunities for systemic improvement in team-based recovery models for mTBI. Despite increasing advocacy for team-based models, no existing studies have systematically synthesized structured interviews across multiple clinical disciplines to inform a conceptual framework grounded in actual practice environments.

### Literature Review

The multifaceted nature of post-concussion symptoms—encompassing physical, cognitive, emotional, and behavioral domains—necessitates a comprehensive and interdisciplinary approach to rehabilitation. While initial management often emphasizes rest and reassurance, individuals with persistent post-concussion symptoms (PPCSs) benefit from more structured and collaborative interventions [3,4].

Mashima et al. [2] advocate for the formation of core interdisciplinary teams (IDTs) comprising speech-language pathologists (SLPs), neuropsychologists, care coordinators, and rehabilitation physicians. They highlight the critical role of consistent interprofessional communication, validation of patient experiences, and consideration of the sociocultural context in recovery. Such interdisciplinary management is posited to reduce fragmented care and promote resilience, functional restoration, and community reintegration.

Building upon this, Nguyen et al. [1] provide empirical support for an interdisciplinary program—i-RECOveR—that integrates psychology, physiotherapy, and medical interventions over a 12-week period. Their study demonstrated that 80% of participants with PPCSs experienced moderate to significant improvements in symptom severity and quality of life. The application of Goal Attainment Scaling (GAS) further underscored the value of individualized, patient-centered outcomes over standard clinical measures.

The importance of shifting from a deficit-based, diagnostic paradigm to a strength-based, functional approach is a recurring theme in the literature [5]. Neuropsychological assessments may not always detect impairments; however, patient-reported cognitive inefficiencies, often exacerbated by sleep disorders, anxiety, depression, and pain, are valid concerns [6,7]. Interdisciplinary collaboration is essential to contextualize these symptoms, validate patient experiences, and tailor interventions to complex biopsychosocial profiles.

Operational pillars of effective interprofessional practice include:Shared Decision-Making: Collaborative goal-setting and motivational interviewing techniques foster autonomy, build trust, and align treatment plans with personal values and life roles [2].Integrated Education and Messaging: Clear, consistent communication across professionals helps prevent “diagnosis threat,” wherein labels like “brain injured” contribute to symptom amplification and prolonged disability beliefs [8].Functional, Goal-Oriented Treatment: The use of dynamic coaching and GAS ensures that cognitive rehabilitation efforts are tied to real-world objectives, promoting the transfer of learning and sustained gains [9].Multimodal Physiotherapy: Physiotherapy targeting vestibular, cervical, and sensorimotor domains has been shown to significantly reduce PPCSs, particularly when delivered alongside psychoeducation and cognitive–behavioral therapy [1].Role Differentiation and Coordination: Successful IDT implementation depends on each discipline contributing its specialized expertise while actively engaging in collaborative problem-solving and coordinated care planning.

Despite the recognized benefits of interdisciplinary approaches, numerous challenges persist in their implementation. A significant barrier is limited access to comprehensive rehabilitation services, leaving many patients and their families with unmet needs [10]. Cognitive and behavioral complications, such as deficits in attention, memory, executive functioning, and impulsivity, further complicate the rehabilitation process, particularly when compounded by co-existing metabolic disorders that may exacerbate cognitive dysfunction [11,12].

Systemic barriers—including fragmented communication channels, differing professional cultures, and logistical constraints—also impede effective interdisciplinary collaboration, leading to disjointed care and suboptimal patient outcomes [13]. Gagnon-Roy et al. (2024) [13] emphasize the necessity of minimal and highly individualized cognitive interventions provided by occupational therapists, reinforcing the need for flexible, tailored strategies within broader team-based frameworks.

Addressing these challenges requires fostering better interprofessional understanding across disciplines. Enhanced interdisciplinary communication, regular cross-disciplinary meetings, collaborative care planning, and structured cross-training can bridge professional silos, promote mutual respect, and ensure cohesive, patient-centered rehabilitation. Such interprofessional synergy is critical to overcoming the identified barriers and improving the overall effectiveness of rehabilitation practices.

In sum, the literature supports a robust, multifaceted, and human-centered model of interdisciplinary care as a promising paradigm for improving recovery outcomes in mTBI. Nevertheless, addressing systemic barriers and cultivating a strong collaborative culture are vital to realizing the full potential of team-based rehabilitation. In the sections that follow, we synthesize data from 94 interviews with diverse rehabilitation professionals to examine how these principles are implemented—and occasionally challenged—in clinical practice.

## 2. Materials and Methods

### 2.1. Interview Sampling and Participant Inclusion

Interview participants were selected through an iterative process that integrated purposive, convenience, and snowball sampling strategies. This recruitment process was initially anchored to hypotheses derived from the Business Model CANVAS framework, as stipulated by the NIH I-Corps program. Early-stage assumptions identified behavioral optometrists as likely early adopters of a visual-cognitive intervention for mild traumatic brain injury (mTBI). However, as interviews progressed, the sample was deliberately diversified to encompass a broader array of allied health professionals—including neuropsychologists, occupational therapists, speech-language pathologists, and rehabilitation physicians—who play critical roles in interdisciplinary mTBI care.

A total of 100 interviews were conducted and also had transcriptions available. Of these, 6 transcripts were excluded from analysis due to insufficient length or lack of substantive content necessary for thematic coding. The final analytic sample thus consisted of 94 participants. Eligibility criteria required that participants hold a graduate-level clinical or research degree (e.g., Ph.D., O.D., M.D., M.S., or equivalent) and possess a minimum of five years of post-licensure or professional practice experience in a relevant healthcare or scientific discipline.

Initial interviewees were identified through the authors’ professional networks and institutional affiliations spanning California, Alabama, New York, Colorado, Massachusetts and Illinois. These individuals subsequently referred colleagues with domain expertise in the diagnosis, treatment, and study of cognitive and sensory sequelae following mTBI. Recruitment was conducted through targeted email outreach, institutional contacts, and NIH I-Corps cohort interactions.

Participants were drawn from a wide variety of clinical settings, including academic medical centers, rehabilitation hospitals, private practices, and community-based health organizations. All participants had direct experience in clinical care or research related to mTBI and were selected for their engagement with interdisciplinary approaches to neurorehabilitation.

### 2.2. Interview Structure and Administration

Interviews were conducted using a structured protocol derived from the I-Corps “customer discovery” framework but tailored for clinical relevance. Interviews were conducted by the lead author and second author. Each interviewee was told that the goal of the interviews was to understand the professional “pain points” in mTBI care delivery, particularly regarding diagnosis, rehabilitation, interdisciplinary communication, and technology adoption. Core questions addressed current treatment practices, challenges in managing cognitive symptoms, interprofessional communication strategies, and the use of technology or tools in patient care. All participants were told about the nature of I-Corps, the reasons for conducting the interviews, and what we hoped to learn by conducting the interviews. The characteristics of the interviewers were disclosed, informing the interviewees that author one was an academic researcher at a southeastern university in the US, and that the second author was a neuroscientist, inventor, and founder of Perception Dynamics Institute. As per ICorps requirements, data was managed in the Air Table CRM software.

Although the interview protocol began as a fixed set of questions, it evolved into a semi-structured format as new insights emerged. This allowed interviewers to pursue emergent themes while maintaining consistency across participants. Interviews were conducted primarily via Zoom, with a minority occurring in person. For interviews conducted in clinical settings or at academic conferences field notes were made in addition to those in the Zoom platform. Each session lasted approximately 30 to 45 min.

### 2.3. Qualitative Analysis and Thematic Synthesis

The qualitative analysis in this study followed a pragmatic, inductive thematic approach designed to extract meaningful patterns across a diverse interdisciplinary stakeholder sample. While initially rooted in constant comparison and grounded theory logic, the analytic strategy was significantly augmented by the integration of structured qualitative software (NVivo 15) and custom-developed Python-based NLP pipelines for advanced semantic mapping and classification.

#### 2.3.1. NVivo-Assisted Structural Coding

Interview transcripts were initially prepared and imported into NVivo 15 for structural coding. NVivo’s autocoding features were applied based on document formatting and speaker tags to segment responses by participant and interview questions. Case classifications were created for each participant, with associated metadata fields such as profession and case ID imported via classification sheets. NVivo was used to assign initial structural nodes and to manage the exploratory coding of common experiential domains.

#### 2.3.2. Inductive Theme Extraction and Manual Coding

From the structured NVivo coding base, subject-level themes were extracted through an iterative manual review of transcript excerpts. These themes reflected recurring patterns across interviews and were initially compiled into a spreadsheet that listed themes by participant case ID. This manual phase emphasized inductive emergence of conceptual categories grounded in participants’ language and contextualized experiences.

#### 2.3.3. NLP-Based Semantic Mapping Using Python 3.11

To bridge subject-level themes with a theory-informed thematic model, a custom Python script was developed using the sentence-transformers library (HuggingFace) and the pandas library. This script performed the following operations:**Data Loading**: It ingested two datasets—one containing subject-level themes, and another containing a curated list of 11 abstracted themes each linked to a broader superordinate category (e.g., “Systemic Barriers”, “Care Infrastructure”).**Sentence Embedding Generation**: Each theme phrase was embedded in high-dimensional semantic space using the all-MiniLM-L6-v2 transformer model to capture deep contextual meaning beyond surface-level lexical similarity.**Similarity Matrix Computation**: Cosine similarity was computed between each subject-level theme and all abstracted themes. Matches above a minimal threshold (0.1) were retained to allow exploratory clustering.**Multi-Match Classification Output**: For each subject theme, the top 11 most semantically similar abstracted themes were identified and annotated with both the match score and their corresponding superordinate theme. These mappings were saved in a structured csv file for integration back into NVivo or export to visualization tools.

This semantic classification pipeline allowed for **data-driven yet theory-anchored mapping** of the inductive interview content onto a defined thematic ontology, minimizing bias in manual categorization while preserving interpretive richness.

#### 2.3.4. Saturation and Final Refinement

Thematic saturation was deemed achieved prior to the completion of all 94 interviews, as minimal novel concepts emerged after the 80th transcript. The final clustering of themes into superordinate categories was reviewed collaboratively among the research team to ensure conceptual validity and intersubjective agreement. These categories included: *Systemic Barriers*, *Care Infrastructure*, *Technology Integration*, *Coordinated Training*, and *Discipline-Specific Expertise (See Figure 1)*.

This hybrid approach—combining NVivo’s robust case management with large-scale semantic similarity analysis—afforded both depth and scalability, aligning with best practices in translational qualitative research under the NIH I-Corps paradigm.

### 2.4. Ethical Considerations

This project was conducted under the auspices of the NIH I-Corps program, which is exempt from IRB oversight due to its focus on translational research and stakeholder discovery rather than human subjects research per se. Nonetheless, all participants provided informed verbal consent at the outset of each interview. Participants were specifically told that the authors were planning to turn the results of the interviews into an article, and verbal assent was obtained. Participants were assured of the voluntary nature of their involvement, and data were anonymized in all reporting and dissemination activities.

## 3. Results

### 3.1. Themes from Interdisciplinary Professional Interviews

A synthesis of 94 professional interviews with rehabilitation experts across disciplines reveals several converging themes regarding the treatment of mTBI. The final dataset included participants representing a diverse array of professions: clinical neuropsychologists, rehabilitation physicians, optometrists (ODs), occupational therapists, speech-language pathologists, physical therapists, audiologists, psychologists, neurologists, nurses, patients, and caregivers. See Table 1 for a full breakdown by profession. The themes that emerged included the following:Shared Commitment to Functional Recovery: Regardless of discipline, providers emphasized restoring daily functioning and life participation as central goals. Treatments were consistently tailored to real-world tasks (e.g., managing a calendar, navigating a classroom, returning to sport), reinforcing the practical, patient-centered nature of mTBI care.Recognition of Cognitive Persistence in mTBI: Most clinicians acknowledged that mTBI can yield lasting cognitive symptoms, particularly in attention, working memory, and executive function. SLPs and neuropsychologists reported a strong correlation between cognitive symptoms and comorbidities such as anxiety, depression, or sensory overload. Optometrists noted a similar persistence of visual symptoms. In contrast, some neurologists and surgeons questioned the duration and clinical significance of post-concussive cognitive complaints.Barriers to Integration: Common frustrations included limited interdisciplinary communication, lack of shared medical records, role ambiguity, and insurance obstacles. Several professionals emphasized that the absence of structured team meetings and coordinated planning led to fragmented care. Others lamented the difficulty of obtaining referrals across disciplines.Discipline-Specific Insights: SLPs detailed their use of evidence-based cognitive-linguistic therapies, often tailored to executive deficits.

#### Optometrists (Behavioral)

Optometrists emphasized neurovisual interventions such as syntonic phototherapy and digital eye-tracking tools.

Figure 2 highlights the distribution of superordinate thematic categories derived exclusively from interviews with optometrists (ODs), who comprised the most frequently sampled profession in this study. The heatmap illustrates the presence and relative intensity of five major theme domains—Care Infrastructure, Coordinated Training, Discipline-Specific Expertise, Systemic Barriers, and Technology Integration—across individual OD respondents.

This subgroup-level analysis reveals that Coordinated Training and Discipline-Specific Expertise were the most consistently and intensely represented themes among ODs. These findings suggest that optometrists, more than many other professionals interviewed, emphasized the need for enhanced collaborative education and role clarification within multidisciplinary TBI care environments. The pronounced intensity under Technology Integration among some respondents also points to a subset of ODs advocating for the more seamless incorporation of diagnostic and therapeutic technologies in their clinical workflow—particularly those relevant to post-concussion syndrome.

Notably, the theme of Systemic Barriers appeared intermittently across cases, indicating that while ODs do encounter institutional and reimbursement challenges, these are less uniformly experienced than professional training gaps. The sparse representation of Care Infrastructure suggests that optometrists may view structural and logistical components of care delivery as secondary to issues of competency, coordination, and technological capacity.

By isolating the OD profession for fine-grained analysis, this figure offers insight into the distinctive concerns and priorities expressed by eye care professionals within the broader constellation of mTBI providers. These data reinforce the conceptual framework’s emphasis on profession-specific entry points into interdisciplinary rehabilitation, and they support targeted outreach strategies for engaging optometrists in future implementation efforts.

Occupational Therapists

Occupational therapists were another important subgroup. OTs highlighted sensory integration, vestibular rehab, and reflex modulation, often applying integrative methods like PATH neurotraining.

Physicians and Surgeons focused on diagnostic imaging and pharmacologic stabilization, generally deferring long-term rehab to allied professionals.

5.Technology, Training, and Advocacy: Participants highlighted the growing role of assistive software, from cognitive training programs to vision-tracking apps. However, adoption was uneven due to cost, training gaps, and interoperability. Many professionals described acting as advocates, guiding patients through complex rehabilitation systems and educating families to improve care continuity.

In summary, the interviews offer real-world confirmation of what the literature suggests: while interprofessional collaboration is essential to the effective treatment of mTBI, it remains hampered by systemic silos, training inconsistencies, and resource constraints. Nonetheless, the interviews also highlight innovation, shared purpose, and opportunities for reform.

### 3.2. Divergent Professional Perspectives on Symptom Persistence and Rehabilitation Roles

While there was broad consensus among professionals on the need for functional, patient-centered recovery approaches, interviews also elucidated notable divergences across disciplines—particularly in how providers conceptualize the persistence and clinical relevance of post-concussive symptoms. These differences, while not universally polarized, reflect underlying variations in professional training, epistemological frameworks, and clinical focus.

Speech-language pathologists (SLPs) and neuropsychologists, for instance, frequently emphasized the enduring nature of cognitive and communicative impairments following mTBI. One SLP observed, “We see patients months later who still can’t follow multistep instructions—this isn’t just anxiety, it’s real processing dysfunction.” A neuropsychologist echoed this sentiment, stating, “Even when standard batteries come back normal, patients report difficulties in executive function that meaningfully affect daily life”.

By contrast, several neurologists and neurosurgeons expressed skepticism regarding the prolonged trajectory of cognitive symptoms, often emphasizing objective findings over patient-reported experiences. One neurologist remarked, “From a neurological standpoint, if the scan is clean and it’s six months out, we start questioning whether the symptoms are neurologically driven or psychologically maintained”.

These differences were not purely academic; they shaped referral patterns, treatment recommendations, and expectations for recovery. Optometrists specializing in neurovisual rehabilitation noted that such skepticism often delayed referrals for visual interventions, despite persistent symptoms like photophobia or tracking deficits. Occupational therapists similarly reported that some patients arrived in therapy only after exhausting pharmacological or diagnostic avenues, by which point functional deterioration had already compounded.

This disciplinary heterogeneity underscores the importance of structured interprofessional dialog and shared decision-making frameworks that validate patient-reported experiences while leveraging diverse clinical expertise. Without such integrative approaches, care risks becoming fragmented or hierarchically biased toward certain paradigms of evidence.

The field of Psychology contains two distinct groups that were interviewed in this project, Clinical Psychologists and Clinical Neuropsychologists. Because of the differences in their roles and perspectives, they were analyzed separately. These excerpts exemplify the themes from the two groups.

#### 3.2.1. Clinical Neuropsychologist Sample Excerpts

R001_Clinical_Neuropsychologist

“Most of the referrals we get are after everything else has been tried—imaging, medication, basic OT. By the time they reach us, the patient is demoralized, and the family is burned out. We end up being the ones to translate symptoms into something meaningful. ‘No, you’re not just lazy. No, it’s not just in your head.’ It’s executive dysfunction. It’s working memory collapse under dual-task load.”

R006_Clinical_Neuropsychologist

“I often explain the concept of cognitive load and recovery plateau to families. It’s not that the patient isn’t trying; it’s that the cognitive reserve is depleted. We can sometimes retrain it, but often we’re guiding them to accept new limits and focus on realistic accommodations.”

#### 3.2.2. Clinical Psychologist Sample Excerpts


**R002_Clinical_Psychologist**


“Our work intersects most clearly when anxiety or trauma emerges post-injury. What I often see are individuals who were high-functioning, suddenly stripped of their identity. And we’re trying to help them rebuild narrative coherence. ‘Who am I now?’ becomes the therapeutic core.”


**R041-Clinical-Psychologist**


“The medical folks focus on symptoms and structure. We look at meaning. A headache isn’t just a headache—it’s the anchor of helplessness. The moment we treat it as a metaphor, rather than a target, we get movement. That’s the psychological leverage.”

These illustrative samples reveal a clear distinction in epistemological frameworks and clinical goals between Clinical Neuropsychologists and Clinical Psychologists. I will now proceed with a full thematic comparison across all files in both groups.

Based on the keyword frequency and uniqueness analysis of the Clinical Neuropsychologist and Clinical Psychologist transcripts, several distinctions and overlaps emerge that illuminate both shared and divergent emphases in professional orientation and narrative style. Below is a detailed synthesis of these similarities and differences, grounded in textual evidence from the interviews.

#### 3.2.3. Similarities Across Clinical Neuropsychologists and Clinical Psychologists

Both professional groups frequently invoked terms such as “cognitive,” “think,” “working,” and “brain,” underscoring their shared concern with cognitive functioning in patients with mTBI. These terms reflect a mutual emphasis on cognitive symptoms, rehabilitation engagement, and psychological processing.

For example, both groups regularly discussed executive dysfunction and attention difficulties, albeit through slightly different clinical lenses. A neuropsychologist stated the following:

“By the time they reach us, the patient is demoralized… it’s working memory collapse under dual-task load.”

Whereas a psychologist framed similar symptoms in narrative terms:

“*They’re not just forgetting tasks; they’ve lost the thread of who they were. The executive issues are part of an identity fracture*.”

This convergence suggests an interdisciplinary appreciation for post-concussive cognitive impairment, though anchored in distinct explanatory models.

#### 3.2.4. Distinctive Themes for Clinical Neuropsychologists

Words unique or more dominant in the neuropsychologist corpus include “*assessment*,” “*brain*,” “*limited recovery*,” and “*referral*.” These may reflect a more frequent engagement with structured assessment tools (e.g., references to standardized batteries) and a conversational teaching style characteristic of neuropsychological feedback sessions.

Neuropsychologists emphasized the following:The translation of neurocognitive impairments into functional terms.A focus on diagnostic precision, often highlighting gaps in referral timing.The reality of neurofunctional deficits even in the absence of imaging findings.

As one neuropsychologist articulated the following:

“*We’re not here to confirm the MRI. We’re here to validate the cognitive experience. That’s often the missing link for patients lost in the system*.”

Another said the following:

“*It’s executive dysfunction, not apathy. But unless someone runs the Stroop or Trails, no one catches it*.”

#### 3.2.5. Distinctive Themes for Clinical Psychologists

Clinical Psychologists, by contrast, leaned heavily into themes of “*family*,” “*military*,” “*therapy*,” “*injury*,” and “*clinical*.” Their discourse included the following:Emphasis on **psychosocial readjustment** post-injury.A **therapeutic narrative reconstruction** approach.Frequent engagement with **military or trauma backgrounds**, especially in identity renegotiation.

One psychologist noted the following:

“*A lot of the veterans we see don’t want to talk about cognition. They want to know who they are now. We build from there*.”

Another observed the following:

“*Families are often the barometer. They’ll say, ‘She’s not the same.’ That’s our starting point*.”

#### 3.2.6. Conclusion: Cognitive Convergence, Epistemological Divergence

In summary, while Clinical Neuropsychologists and Clinical Psychologists both confront the enduring impact of mTBI on cognition and selfhood, they diverge in their methods of interpretation and intervention. Neuropsychologists tend toward diagnostic articulation and functional quantification, while psychologists often focus on psychological coherence and emotional processing. These complementary perspectives suggest an opportunity for greater integration—where quantitative neurocognitive profiles inform narrative therapy and, conversely, where emotional insights help refine neuropsychological treatment planning.

### 3.3. Equity, Access, and Structural Determinants of Interdisciplinary Care

While this study focused on the perspectives of clinicians, several participants independently raised concerns about disparities in access to interdisciplinary care. These disparities—rooted in insurance coverage, geographic proximity, and systemic inequities—pose substantial obstacles to the implementation of integrated rehabilitation frameworks.

Clinicians practicing in rural or underserved areas described limited access to allied professionals such as neuro-optometrists or SLPs trained in cognitive-communication therapy. One case manager in a midwestern community noted, “Our patients may have to drive three hours just to see a vestibular therapist, and that’s if their insurance even covers it.” Others lamented the logistical hurdles that prevent marginalized patients from participating in comprehensive team-based care, including transportation barriers, employment constraints, and caregiver burden.

These observations echo findings in the broader literature on health disparities in brain injury care. Research has shown that individuals from lower socioeconomic backgrounds and racial/ethnic minority groups are less likely to receive post-acute rehabilitation services, and when they do, the services tend to be shorter in duration and less comprehensive [14,15]. Additionally, structural racism in healthcare delivery—manifested through implicit bias, diagnostic overshadowing, and underrepresentation of minoritized clinicians—may contribute to differential treatment recommendations and patient engagement in rehabilitation settings.

To address these inequities, future models of interdisciplinary care must include strategies for equitable implementation. This includes expanding tele-rehabilitation networks, advocating for policy reform around Medicaid reimbursement, and embedding cultural competence training in interdisciplinary education. Ultimately, any scalable model must attend not only to the composition of the care team but also to the accessibility of that team across diverse populations.

Collectively, the findings from these 94 professional interviews converge around several interlocking domains critical to successful mTBI rehabilitation. These include discipline-specific expertise, coordinated training structures, integrated documentation and communication systems, and technology-enabled solutions, all shaped by the persistent influence of systemic barriers. Figure 3 presents a conceptual framework synthesizing these domains and their interactions to illustrate the pathway toward successful recovery.

This model illustrates the proposed conceptual framework for interdisciplinary care in mild traumatic brain injury (mTBI) rehabilitation, derived from a synthesis of literature and 94 professional interviews. The framework begins with the TBI patient and emphasizes the importance of discipline-specific contributions, which feed into ongoing training and education. These educational structures support the development of care coordination infrastructure, which in turn enables technology integration to enhance communication, monitoring, and personalized interventions. Systemic barriers, including fragmented communication, reimbursement limitations, and role ambiguity, act as central impediments across all domains. Ultimately, the framework aims to support successful recovery by fostering dynamic, patient-centered, and technologically supported interprofessional collaboration.

## 4. Implications and Future Directions

The findings from this qualitative analysis provide several practical and conceptual implications for improving interdisciplinary care for patients with mTBI.

First, rehabilitation systems should prioritize the integration of care teams at both the structural and procedural levels. Interdisciplinary care models function best when supported by shared documentation systems, routine cross-specialty case reviews, and clearly defined communication protocols. Health systems can benefit from embedding care coordination roles or case managers whose primary responsibility is to facilitate interprofessional collaboration.

Second, there is a strong need for training programs to enhance interdisciplinary competence among healthcare providers. Many participants reported learning about collaboration on the job, often through trial and error. Introducing interprofessional modules into graduate-level clinical education for SLPs, OTs, PTs, optometrists, and medical doctors would address these training gaps and promote earlier exposure to team-based thinking.

Third, policy advocacy and reimbursement reform are essential. The fragmentation described by interviewees often stems from insurance limitations that disincentivize interdisciplinary consultations or undervalue non-physician services, such as neuro-optometric and recreational therapies. Policymakers and professional organizations should advocate for bundled or team-based reimbursement models that reflect the real-world needs of mTBI patients.

Fourth, the use of technology in interdisciplinary mTBI care deserves greater attention and standardization. While some professionals reported success with digital vision tracking, cognitive software, and virtual collaboration tools, adoption remains uneven. Clinical settings would benefit from curated guidelines that identify evidence-based digital tools and describe protocols for their implementation across disciplines.

Finally, future research should explore the longitudinal effects of collaborative care models on both clinical outcomes and cost-effectiveness. Quantitative studies measuring functional recovery, return-to-work rates, and quality of life following interdisciplinary versus siloed care would provide a robust evidence base for health systems aiming to redesign post-concussion services.

In sum, while challenges to integration persist, the voices of these 94 professionals point clearly toward a model of care that is not only collaborative in theory, but also flexible, patient-centered, and dynamically responsive to the evolving needs of individuals recovering from mTBI.

## 5. Conclusions

This study underscores the critical importance of interprofessional collaboration in the effective treatment of mild traumatic brain injury. The literature demonstrates clear advantages to a coordinated, patient-centered approach, particularly when dealing with persistent post-concussion symptoms. Insights from 94 diverse healthcare professionals reinforce this consensus, while also illuminating the ongoing challenges of fragmented systems, discipline-specific silos, and variable training in collaborative practices.

Collectively, the findings suggest that interdisciplinary care for mTBI must evolve beyond conceptual endorsement toward structural implementation. This includes adopting shared electronic health records, embedding interdisciplinary education within professional training pipelines, and advocating for insurance reform that supports coordinated care delivery.

Importantly, the voices of clinicians across neurology, optometry, speech pathology, and rehabilitation provide both validation and urgency to this call for reform. Their perspectives reveal a professional community eager to collaborate but constrained by outdated systems. Moving forward, research must rigorously evaluate the impact of interdisciplinary care models on long-term patient outcomes, while clinicians and institutions work to normalize collaborative frameworks in both acute and post-acute care settings.

Interdisciplinary rehabilitation for mTBI is no longer aspirational—it is essential. As the burden of concussion continues to grow, it is incumbent on the healthcare community to develop and sustain systems that reflect the complexity of recovery and the diversity of professional expertise required to support it.

## Figures and Tables

**Figure 1 medsci-13-00082-f001:**
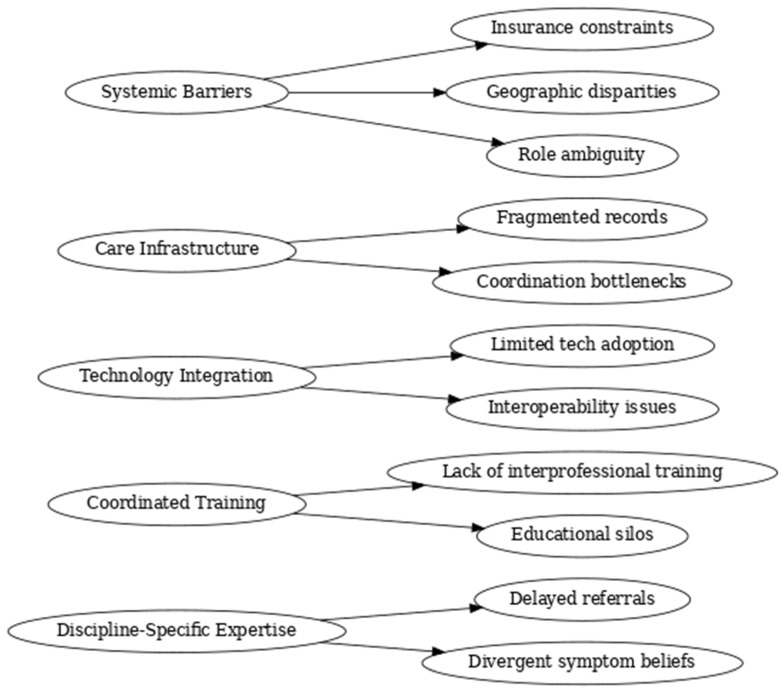
Coding tree derived from thematic analysis of stakeholder interviews. This diagram illustrates the hierarchical structure of themes developed through inductive coding and constant comparison of 94 stakeholder interviews conducted as part of the NIH I-Corps program. Five core conceptual domains emerged: Discipline-Specific Expertise, Coordinated Training, Technology Integration, Care Infrastructure, and Systemic Barriers. Each domain is linked to specific subthemes reflecting clinician-reported experiences and challenges in the diagnosis, treatment, and coordination of mild traumatic brain injury (mTBI) rehabilitation. The coding tree reflects the analytic foundation of the study’s practice-informed conceptual framework for interprofessional care.

**Figure 2 medsci-13-00082-f002:**
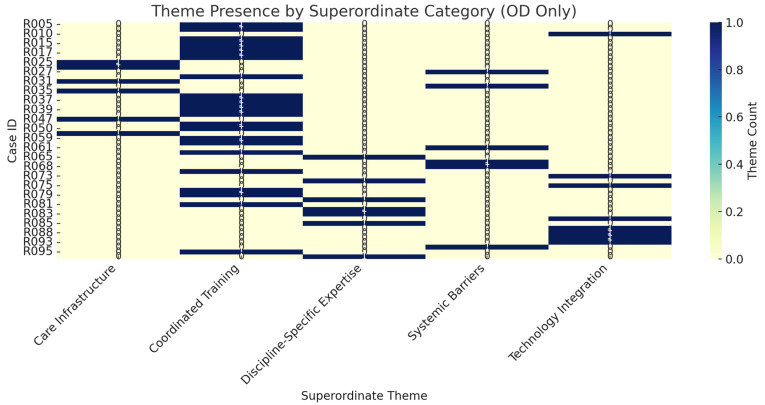
Heatmap of superordinate theme coverage among optometrists (ODs). This figure displays the distribution of superordinate themes identified in interviews with professionals from the OD (Optometrist) category. Because ODs represented the most frequently sampled profession in the study, this subgroup permitted more fine-grained thematic analysis. The intensity of each cell reflects the frequency with which subject-level themes aligned with each superordinate category, as determined through sentence-transformer-based semantic similarity matching.

**Figure 3 medsci-13-00082-f003:**
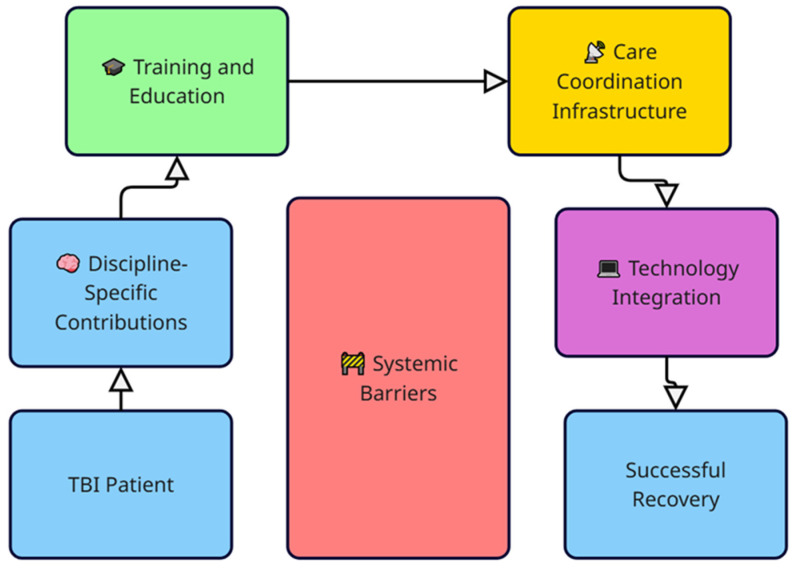
Conceptual framework for interprofessional rehabilitation in mTBI.

**Table 1 medsci-13-00082-t001:** Number of interviewees by profession.

Profession	Number Interviewed
OD	49
Clinical Neuropsychologist	9
OT	7
SLP	6
Nurse	6
Vision Therapist	4
Clinical Psychologist	3
Software Developer	3
Rehabilitation Psychologist	2
Rehabilitation Counselor	2
Reading Specialist	2
Parent	2
Special Education	1
Recreation Therapist	1
Rehabilitation Medicine	1
Physical Therapist	1
Internal Medicine	1
Patient	1
PMR	1
Neurologist	1
Neurosurgeon	1
Total	104

OD = Doctor of Optometry (optometrist); SLP = Speech-Language Pathologist; OT = Occupational Therapist; PT = Physical Therapist. Note: Vision Therapist refers to a para-professional or assistant-level provider without doctoral-level training, included in this study for their clinical involvement in mTBI rehabilitation.

## Data Availability

Deindividuated data available on request from the corresponding author.

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
