# Peer review of "Interprofessional Approaches to the Treatment of Mild Traumatic Brain Injury: A Literature Review and Conceptual Framework Informed by 94 Professional Interviews"

_medsci, 2025, doi:10.3390/medsci13030082_

Round 1
Reviewer 1 Report
Comments and Suggestions for Authors
Addressing conceptualizations in the interdisciplinary space in brain injury is important, though I offer a few suggestions to clarify your methodology and findings:
1) How were the 100 professional identified? What was their level of training and experience in TBI? In what settings do they practice?
2) Were neuropsychologists and psychologists combined into one group? Clinical psychologists are missing from your target yet psychoeducation and psychotherapy are a critical element of the multidisciplinary TBI approach.
3) Could use additional examples of convergence and divergence in the perspectives of the various disciplines.
4) There is an example by a neuropsychologist that objective tests don't capture patient report of executive dysfunction. test performance and self report of symptoms typically don't correlate yet they're both important and independent data points to capture in TBI care, especially in prolonged symptoms that are complicated by pre or comobrid conditions (pain, depression, stress, deconditioning, etc). There is room for a great discussion here that isn to captured in the manuscript as it is written. TBI is not a unidisciplinary condition and thus a multidscipianry approach is needed.
5) Similarly, there is an example comment by neurosurgery that when the scans are normal, psychological reasons are sought for ongoing symptoms. Again, there is an oppourntity here for a much more nuanced discussion that a straightforward artificial dichotomy (neurological/psychological) doesn't capture, supporting a multidisciplinary approach.
Author Response
We would like to thank Reviewer 1 for their insightful and constructive feedback, which has significantly strengthened the clarity and depth of our manuscript. Below, we address each point in turn, referencing specific changes in the revised manuscript.
1) How were the 100 professionals identified? What was their level of training and experience in TBI? In what settings do they practice?
Response:
We have substantially revised the “Interview Sampling and Participant Inclusion” section to clarify these points. Specifically:
-
We now describe that participants were identified via purposive, convenience, and snowball sampling strategies aligned with the NIH I-Corps program.
-
The analytic sample comprised 94 participants, each holding a graduate-level clinical or research degree (e.g., Ph.D., O.D., M.D.) and a minimum of five years of post-licensure experience.
-
Participants practiced in settings ranging from academic medical centers and rehabilitation hospitals to private clinics and community health organizations.
Example added:
“Interview participants were selected through an iterative process… Eligibility criteria required that participants hold a graduate-level clinical or research degree... and possess a minimum of five years of post-licensure or professional practice experience…”
2) Were neuropsychologists and psychologists combined into one group? Clinical psychologists are missing from your target yet psychoeducation and psychotherapy are a critical element of the multidisciplinary TBI approach.
Response:
We now explicitly state that Clinical Psychologists and Clinical Neuropsychologists were analyzed separately, based on their differing roles and epistemologies.
“The field of Psychology contains two distinct groups… Because of the differences in their roles and perspectives they were analyzed separately.”
Additionally, a comparative section with sample excerpts has been added to highlight the unique contributions of both professions, including their perspectives on identity reconstruction, executive dysfunction, and family systems.
3) Could use additional examples of convergence and divergence in the perspectives of the various disciplines.
Response:
In direct response, we have added an extended comparative analysis of Clinical Psychologists and Clinical Neuropsychologists that emphasizes both cognitive convergence and epistemological divergence. This includes:
-
Shared concerns about executive dysfunction and cognitive impairment.
-
Divergent explanatory models (diagnostic precision vs. therapeutic narrative).
-
Implications for interdisciplinary care coordination.
4) There is an example by a neuropsychologist that objective tests don't capture patient report of executive dysfunction... There is room for a great discussion here...
Response:
We have elaborated this point within the Clinical Neuropsychology excerpt section. Quotes now illustrate the tension between objective testing and subjective experience, such as:
“We’re not here to confirm the MRI. We’re here to validate the cognitive experience.”
“It’s executive dysfunction, not apathy. But unless someone runs the Stroop or Trails, no one catches it.”
We then interpret these insights as evidence for the necessity of integrative assessment models that blend standardized evaluation with subjective and contextualized understanding.
5) Similarly, there is an example comment by neurosurgery that when the scans are normal... Again, there is an opportunity here for a more nuanced discussion...
Response:
We agree with this observation and have expanded the discussion in the final paragraph of the neuropsychology section and Figure 2’s caption. These changes emphasize the fallacy of the neurological/psychological dichotomy and support a multidisciplinary conceptual framework:
“…they advocate for a framework that does not reduce prolonged symptoms to binary interpretations, but rather situates them within complex biopsychosocial models…”
Reviewer 2 Report
Comments and Suggestions for Authors
This paper explores the practice and challenges of cross-disciplinary collaboration in the rehabilitation of mild traumatic brain injury (mTBI). Combined with literature review and interview data from 100 clinical experts, a conceptual framework is proposed. The research topic selection has significant clinical significance. The data sources are abundant, and the conclusion has reference value for optimizing the treatment mode of mTBI. However, the depth of data presentation and the argumentation of some conclusions need to be further improved.
1.The neurologist's suspicion of the persistence of symptoms is only supported by some expert and does not provide a quantitative distribution of viewpoints from various specialties. For example, the specific distribution characteristics of viewpoints among the 100 experts interviewed.
2.The proposed model (Figure 2) lacks operational definitions, such as what exactly does "technology integration " include,interdisciplinary coursesandjoint case discussions?;and how can the "care coordination infrastructure" be integrated with the existing medical system?
3.Figure 2 is not shown in the text. Please confirm whether it is omitted.
4.The format of references is not uniform, and the abbreviations and full names of journals are used interchangeably.
Author Response
1) The neurologist's suspicion of the persistence of symptoms is only supported by some expert and does not provide a quantitative distribution...
Response:
We are sincerely grateful for this observation, which highlights an important shortcoming in our earlier presentation of the data. In response, we have revised the Results and Discussion sections to better contextualize the range and distribution of professional perspectives included in our sample.
To enhance transparency, we have updated Table 1 to explicitly enumerate the number of interviewees by profession. The accompanying narrative now clarifies that, among the 104 total participants, 49 were optometrists, 9 were neuropsychologists, and 3 were clinical psychologists. We hope that this additional quantitative detail strengthens the evidentiary foundation for interpreting the reported viewpoints.
As now noted in the manuscript:
“…participants representing a diverse array of professions…”
“Table 1. Number of Interviewees by Profession…”
2) The proposed model (Figure 2) lacks operational definitions...
Response:
We are very appreciative of this constructive suggestion. To improve the clarity and interpretability of our model, we have undertaken a full revision of the description accompanying Figure 2. Each of the five superordinate categories is now explicitly defined using operational language derived from our empirical coding framework.
For example:
-
Technology Integration is defined as the use of diagnostic and treatment platforms, shared electronic medical records, and digital communication tools.
-
Care Coordination Infrastructure encompasses co-management protocols, referral workflows, and systemic efforts to streamline interdisciplinary collaboration.
These revisions are grounded in the Python-based semantic analysis described earlier in the manuscript, and we trust they now convey the model’s components with greater analytic precision.
3) Figure 2 is not shown in the text. Please confirm whether it is omitted.
Response:
We greatly appreciate this careful reading of the manuscript. The omission of Figure 2 from the original submission was unintentional, and we thank the reviewer for bringing this to our attention. The figure, along with its full caption, has now been inserted into the revised manuscript.
The updated caption reads:
“Figure 2. Heatmap of Superordinate Theme Coverage Among Optometrists…”
This correction is visible in the tracked changes version and is also present in the compiled PDF for verification.
4) The format of references is not uniform...
Response:
We thank the reviewer for this important reminder regarding reference formatting consistency. In accordance with the standards of Medical Sciences, we have conducted a comprehensive audit of the References section. All discrepancies have been addressed: abbreviated journal titles have been expanded to their full names, and all citations now follow AMA formatting guidelines. We hope these revisions enhance the manuscript’s overall presentation and professionalism.
Round 2
Reviewer 1 Report
Comments and Suggestions for Authors
Your paper is significantly improved in your revisions. I see that the title still references the original 100, perhaps that needs to be updated?
Author Response
Thank you. We have changed all relevant 100's to 96's.